# The epidemiology of falls in Portugal: An analysis of hospital admission data

Filipa Sampaio [1]*, Paulo Nogueira [2,3,4,5], Raquel Ascenção [3,6,7], Adriana Henriques [2,5], Andreia Costa [2,5]

1 Department of Public Health and Caring Sciences, Uppsala University, Uppsala, Sweden, 2 Instituto de Saúde Ambiental (ISAMB), Faculdade de Medicina, Universidade de Lisboa, Lisboa, Portugal, 3 Instituto de Medicina Preventiva e Saúde Pública, Faculdade de Medicina, Universidade de Lisboa, Lisboa, Portugal, 4 Área Disciplinar Autónoma da Bioestatística (Laboratório de Biomatemática), Faculdade de Medicina, Universidade de Lisboa, Lisboa, Portugal, 5 Nursing Research, Innovation and Development Centre of Lisbon (CIDNUR), Nursing School of Lisbon, Lisbon, Portugal, 6 Centro de Estudos de Medicina Baseada na Evidência, Faculdade de Medicina, Universidade de Lisboa, Lisboa, Portugal, 7 Escola Superior de Tecnologia da Saúde, Instituto Politécnico de Lisboa, Lisbon, Portugal

* filipa.sampaio@pubcare.uu.se

## Abstract

### Background

Falls are a common cause of injury and pose an increased risk of morbidity, mortality, and lifelong disability. Falls encompass a troublesome definition and can pose challenges in epidemiological studies. Data on fall-related hospital admissions in Portugal remain unpublished. This study aimed to examine the epidemiology of fall-related hospital admissions in the Portuguese population between 2010 and 2018. It also aimed to examine annual rates of fall-related hospital admissions using three methodological approaches.

### Methods

The Portuguese Hospital Morbidity Database was used to identify all cases resulting in one or more inpatient admission in public hospitals related to falls from 2010 to 2018. Fall-related hospital admissions were described by age groups, sex, geographical area of residence, and type of fall. Annual rates were computed using three approaches: i) based on the number of inpatient admissions with an ICD code of fall, ii) based on the number of patients admitted to inpatient care with an ICD code of fall, and iii) based on the number of inpatient admissions with a principal diagnosis of injury.

### Results

Between 2010 and 2018, 383,016 fall-related admissions occurred in 344,728 patients, corresponding to 2.1% of the total number of hospitalizations during the same period. Higher rates were seen among the younger (20–25) and the oldest age groups (+85), males until the age of 60, females from the age of 60, and areas of residence with a higher aging index. An overall rate of falls per 100,000 population was estimated at 414 (based on number of admissions), 373 (based on number of patients) and 353 (based on number of admissions with a principal diagnosis of injury).

**Data Availability Statement:** All available data are within the manuscript and its Supporting information file. This study was based on the analysis of proprietary data, owned by the Portuguese Central Administration of the Health

System, I.P. (ACSS). Aggregated data can be accessed via the Transparency Portal held by the Portuguese Ministry of Health, at the following link: https://transparencia.sns.gov.pt/explore/?sort= title&q=morbilidade+hospital.

**Funding:** The author Paulo Nogueira received computational support of INCD – Instituto Nacional de Computação Distribuida – which is funded by FCT and FEDER under the project 01/SAICT/2016 No. 022153. The funders had no role in study design, data collection and analysis, decision to publish, or preparation of the manuscript.

**Competing interests:** The authors have declared that no competing interests exist.

## Conclusions

This study provides an overall picture of the landscape of falls in a scarcely explored setting. The results aim to contribute to identifying appropriate preventive interventions and policies for these populations.

## Introduction

Falls are one of the most common causes of injury and pose an increased risk of morbidity, mortality, and lifelong disability, particularly in the older age groups [1].

The latest Global Burden of Diseases, Injuries, and Risk Factors Study (GBD, 2019) showed falls ranked as the 21st leading cause of disability-adjusted life years in 2019 [2]. In the oldest age group (75+ years) falls ranked 8th, outranking conditions such as chronic kidney disease, prostate cancer, or road injuries. Because of this, most research on the risk factors for falls and fall-related injuries is focused on older adults [3].

Regardless of age, most falls result from the interplay between predisposing factors (intrinsic or extrinsic) and precipitating factors in a person's environment [4].

The definition of falls in the literature has lacked standardisation, due to the methods utilized for identifying when a fall has occurred, methods of analysis, and details reported. This heterogeneity makes it difficult to aggregate or compare data from different studies. Moreover, falls can have a variety of outcomes ranging from no injury to life-threatening consequences. Nevertheless, several studies have contributed to the knowledge of the epidemiology of falls in Portugal. The GBD 2019 estimates the age-standardized incidence rate of falls that require medical care at 3,112 (95% uncertainty interval 2,695 to 3,627) per 100,000 [1].

In Portugal, the proportion of a self-reported fall in the previous 12 months has been previously estimated at 24.1% in community-dwelling adults in the EpiReumaPt study [5]. Results from this population-based cross-sectional study show an increasing association between age and falls, with people aged 75 and older having greater odds of falling (OR = 1.86; 95% CI 1.49–2.31). Neurologic and rheumatic diseases were also significantly associated with falls.

National data on emergency hospital admissions due to falls are also available through EVITA's injury surveillance system. Injuries caused by falls at home were the most frequently recorded, with young (0 to 14 years) and older (65+) patients being the most affected [6]. In 2018, 30,196 falls among people aged 65 and older were recorded in the EVITA system. The majority of falls occurred at home (64.4%), followed by public areas (9.7%) and outdoor spaces (9.0%); these differences were statistically significant (p<0.01). The type of lesion most frequently observed was bruise (64.1%) and open wound (14.9%) [7].

Falls encompass a troublesome definition and can pose challenges in epidemiological studies. Data on fall-related hospital admissions in Portugal remain unpublished to the best of our knowledge. This study aimed to examine the epidemiology of fall-related hospital admissions in the Portuguese population between 2010 and 2018. It also aimed to examine the rate of fall-related hospital admissions, using three methodological approaches.

By exploring the landscape of fall-related admissions over the years, we seek to identify trends and possible unmet needs, thus tackling the existing knowledge gap in this matter, contributing to deploying appropriate cost-effective preventive interventions and policies for these populations.

## Methods

### Data sources and case identification

This study used a descriptive design to examine the epidemiology of fall-related hospital admissions in the Portuguese population, between 2010 and 2018. The Portuguese Hospital Morbidity Database was used for this research. The Hospital Morbidity Database is a national registry of all public hospital-related care, both inpatient and ambulatory care, centrally held by the Central Administration of the Health System (ACSS). Around 70% of all inpatient hospital admissions occur in public hospitals [8]. In 2017, Portugal registered a population of 10,291,027 inhabitants, 53% of these women [9]. The Portuguese health system includes three overlapping systems. All residents have access to the National Health Service (NHS), which is universal and almost free. The NHS is centrally financed by the Ministry of Health, funded mainly through general taxation. In addition, special health insurance schemes cover specific sectors or professions, and can be either public or private. Private Voluntary Health Insurance is supplementary and speeds up access to elective hospital treatment and ambulatory consultations; it also increases the choice of provider. Between one-fifth and one-fourth of the population has a second health insurance coverage through special schemes or private insurance. Health care is delivered by both public and private providers, with public provision being predominant in primary care and hospital care [10]. Being primarily created for administrative purposes, the Hospital Morbidity Database comprises data on patient demographics, admission and discharge information (including responsible hospital, date and time), in-hospital mortality, International Classification of Diseases (ICD) diagnosis codes, and diagnostic related grouping codes (for cost information). Therefore, clinical data not coded through ICD is absent. All records for individuals are linked utilizing an encrypted patient identification code.

This study included all inpatient admissions related to falls registered between 2010 and 2018. Inpatient admissions related to falls were defined as inpatient episodes with a diagnosis of falls. Falls were defined following the ICD 9th Revision, Clinical Modification (ICD-9-CM) codes E-880-888 and E829.3 and the 10th revision, Clinical Modification/Procedure Coding System (ICD-10-CM/PCS) codes W00-W19. Further analyses considered inpatient episodes with a diagnosis of falls and injury. Injuries were defined following the ICD 9th Revision (ICD-9) codes 800–848, 850–854, 860–887, 890–897, 900–959, and the 10th revision (ICD-10) codes S00-S99, T07 and T14, T15-28, T30-34, and T36-78. For each episode, data related to the type of falls or further characterization was only possible trough coded ICD-9-CM or ICD-10-CM.

### Statistical analyses

This study was exploratory and aimed at investigating the distribution of falls in the whole inpatient population. Therefore, we had no a priori hypothesis to test, as this requires previous knowledge of the phenomenon in the Portuguese setting.

The total number of inpatient admissions related to falls was summarized by age groups, sex, geographical area of residence, and type of fall.

Rates of fall-related hospital admissions were computed using three different approaches: i) based on the total number of inpatient admissions with an ICD code of fall, ii) based on the total number of patients admitted to inpatient care with an ICD code of fall, and iii) based on the total number of inpatient admissions with a principal diagnosis of injury. We estimated annual rates, between 2010 and 2018, for each approach.

Data were analyzed using R version 3.6.3 and R studio version 1.2.5001 [11].

## Results

### Total inpatient admissions related to falls

Between 2010 and 2018, 383,016 inpatient admissions related to falls occurred in 344,728 patients. This number corresponded to 2.1% of the total number of admissions during the same period. In Table 1, the annual total number of patients admitted to inpatient care related to falls, as well as total and mean number of admissions for the same period are reported. The mean annual number of admissions related to falls over this period remained somewhat stable and ranged between 1.06 and 1.19 per patient.

Fig 1 shows the distribution of the mean number of admissions per patient over the analysis period. The mean number of fall-related admissions appears higher among the younger age groups (20–25) and the oldest age groups (+85).

### Rates of fall-related hospital admissions

The estimated overall rate of falls, per 100,000 population, for the analysis period was 414 (based on the total number of episodes), 373 (based on the total number of individuals) and 353 (based on the total number of episodes with a principal diagnosis of injury). Fig 2 shows an overall steady increase in the rates of falls, from 2010 to 2016, wherefrom a sharp decline can be observed, for both males and females, both when taking into account episodes and patients.

**Table 1. Number of admissions related to falls, between 2010 and 2018, in the Portuguese population.**

|  | 2010 | 2011 | 2012 | 2013 | 2014 | 2015 | 2016 | 2017 | 2018 | Total |
|---|---|---|---|---|---|---|---|---|---|---|
| Patients | 33,835 | 36,514 | 36,742 | 39,456 | 39,359 | 41,044 | 43,658 | 39,270 | 34,850 | 344,728* |
| Admissions | 38,288 | 43,592 | 41,578 | 44,173 | 43,788 | 45,117 | 47,193 | 42,042 | 37,245 | 383,016 |
| Mean number of admissions | 1.13 | 1.19 | 1.13 | 1.12 | 1.11 | 1.09 | 1.08 | 1.07 | 1.06 | 1.11 |

*The total does not correspond to the sum of patients over the full period, as some patients appear more than once, and in more than one year.

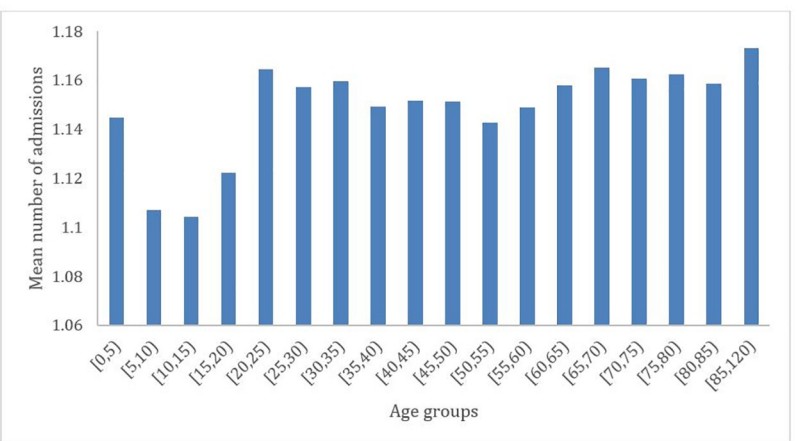

**Fig 1. Mean number of admissions per patient related to falls, between 2010 and 2018, in the Portuguese population.**

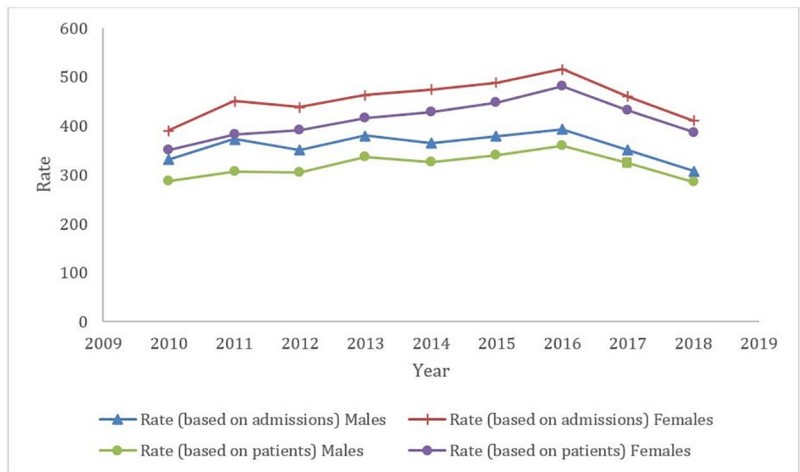

**Fig 2. Rates of falls (per 100,000 population at risk) by sex, based on admissions and based on patients, between 2010 and 2018, in the Portuguese population.**

## Admissions related to falls by age and sex

Table 2 shows the total number and rates of inpatient admissions related to falls by age and sex, between 2010 and 2018. In both scenarios, males show higher rates of fall-related admissions than females until the age of 60. However, from the age of 60 years, females represent the largest proportion of fall-related admissions. Annual rates of fall-related admissions by sex are presented in S1 Table, and the number of patients and admissions related to falls by age group are presented in S2 Table.

The distribution of fall-related admission rates by geographical area of residence is shown in Table 3. The Centre region of Portugal registered the highest rate of fall-related admissions over the years, with an overall rate ranging between 411.6 (based on the number of patients) and 487.31 (based on the number of admissions) per 100,000 population. The islands of Madeira and Azores (not displayed in Table 3) registered rates between 87.8 (based on the number of patients) and 179.19 (based on the number of admissions), and between 82.83 (based on the number of admissions) and 92.7 (based on the number of patients), respectively. Areas of residence with a higher aging index presented the highest rates.

## Number of admissions related to falls by type of fall

Table 4 describes the number of inpatient admissions related to falls, by type of fall. Across all years, the most common type of fall was "Other and unspecified fall" (ranging between 71.51% and 80.22%) followed by "Accidental fall on same level from slipping, tripping or stumbling" (ranging between 5.16% and 10.01%).

## Discussion

This study aimed to examine the epidemiology of fall-related hospital admissions in the Portuguese population, from 2010 to 2018. It also aimed to estimate the rate of fall-related admissions, using three methodological approaches.

The analysis of the Hospital Morbidity Database found that, between 2010 and 2018, 383,016 inpatient admissions related to falls occurred in 344,728 patients. Males registered higher rates of fall-related admissions than females until the age of 60. From the age of 60,

**Table 2. Distribution of rates of falls (per 100,000 population at risk) based on patients and admissions related to falls, between 2010 and 2018, in the Portuguese population.**

| Age groups (years) | Population | | Number of admissions | | | Rate (based on number of admissions)* | | | Number of patients | | | Rate (based on number of patients)* | | |
|---|---|---|---|---|---|---|---|---|---|---|---|---|---|---|
| | M | F | M | F | Total | M | F | Total | M | F | Total | M | F | Total |
| 0–5 | 2,099,393 | 2,004,989 | 3394 | 2211 | 5605 | 161.7 | 110.3 | 136.6 | 2948 | 1949 | 4897 | 140.4 | 97.2 | 119.3 |
| 5–10 | 2,308,309 | 2,202,375 | 4712 | 2604 | 7316 | 204.1 | 118.2 | 162.2 | 4260 | 2350 | 6610 | 184.6 | 106.7 | 146.5 |
| 10–15 | 2,483,506 | 2,366,507 | 5264 | 1933 | 7197 | 212.0 | 81.7 | 148.4 | 4763 | 1756 | 6519 | 191.8 | 74.2 | 134.4 |
| 15–20 | 2,555,769 | 2,448,497 | 4271 | 1146 | 5417 | 167.1 | 46.8 | 108.2 | 3806 | 1022 | 4828 | 148.9 | 41.7 | 96.5 |
| 20–25 | 2,531,072 | 2,481,552 | 3728 | 1097 | 4825 | 147.3 | 44.2 | 96.3 | 3194 | 951 | 4145 | 126.2 | 38.3 | 82.7 |
| 25–30 | 2,622,834 | 2,642,593 | 4027 | 1295 | 5322 | 153.5 | 49.0 | 101.1 | 3475 | 1125 | 4600 | 132.5 | 42.6 | 87.4 |
| 30–35 | 2,942,058 | 3,087,519 | 4884 | 1708 | 6592 | 166.0 | 55.3 | 109.3 | 4177 | 1508 | 5685 | 142.0 | 48.8 | 94.3 |
| 35–40 | 3,364,706 | 3,603,756 | 6223 | 2419 | 8642 | 184.9 | 67.1 | 124.0 | 5330 | 2191 | 7521 | 158.4 | 60.8 | 107.9 |
| 40–45 | 3,449,588 | 3,710,444 | 7194 | 3209 | 10,403 | 208.5 | 86.5 | 145.3 | 6165 | 2868 | 9033 | 178.7 | 77.3 | 126.2 |
| 45–50 | 3,316,279 | 3,595,950 | 8354 | 4365 | 12,719 | 251.9 | 121.4 | 184.0 | 7163 | 3886 | 11,049 | 216.0 | 108.1 | 159.8 |
| 50–55 | 3,210,570 | 3,517,355 | 9234 | 6653 | 15,887 | 287.6 | 189.1 | 236.1 | 7972 | 5934 | 13,906 | 248.3 | 168.7 | 206.7 |
| 55–60 | 2,975,582 | 3,309,496 | 9925 | 9155 | 19,080 | 333.5 | 276.6 | 303.6 | 8534 | 8073 | 16,607 | 286.8 | 243.9 | 264.2 |
| 60–65 | 2,724,730 | 3,101,002 | 10,300 | 11,905 | 22,205 | 378.0 | 383.9 | 381.2 | 8815 | 10,363 | 19,178 | 323.5 | 334.2 | 329.2 |
| 65–70 | 2,392,718 | 2,852,586 | 11,186 | 15,784 | 26,970 | 467.5 | 553.3 | 514.2 | 9465 | 13,684 | 23,149 | 395.6 | 479.7 | 441.3 |
| 70–75 | 1,971,213 | 2,518,396 | 12,614 | 21,713 | 34,327 | 639.9 | 862.2 | 764.6 | 10,743 | 18,834 | 29,577 | 545.0 | 747.9 | 658.8 |
| 75–80 | 1,620,271 | 2,264,592 | 15,729 | 32,277 | 48,006 | 970.8 | 1425.3 | 1235.7 | 13,497 | 27,806 | 41,303 | 833.0 | 1227.9 | 1063.2 |
| 80–85 | 1,135,065 | 1,823,388 | 17,057 | 41,700 | 58,757 | 1502.7 | 2287.0 | 1986.1 | 14,794 | 35,928 | 50,722 | 1303.4 | 1970.4 | 1714.5 |
| ≥85 | 761,723 | 1,626,695 | 21,517 | 62,225 | 83,742 | 2824.8 | 3825.2 | 3506.2 | 18,512 | 52,864 | 71,376 | 2430.3 | 3249.8 | 2988.4 |

F: Females; M: Males.

*Rate per 100,000 population at risk (estimated resident population for Portugal).

**Table 3. Distribution of rates of falls (per 100,000 population at risk) by geographical area of residence, between 2010 and 2018, in the Portuguese population.**

| Area of residence | Admissions | Population | Rate (based on number of admissions) | Aging index 2015 | Patients | Rate (based on number of patients) |
|---|---|---|---|---|---|---|
| Alentejo | 29,735 | 6,598,373 | 450.64 | 191.6 | 26,541 | 402.2 |
| Algarve | 16,013 | 3,987,539 | 401.58 | 138.4 | 14,019 | 351.6 |
| Lisbon | 96,738 | 25,398,930 | 380.87 | 131.7 | 79,317 | 312.3 |
| Centre | 99,607 | 20,440,118 | 487.31 | 183.3 | 84,137 | 411.6 |
| North | 130,948 | 32,650,164 | 401.06 | 139.5 | 113,745 | 348.4 |

females represented the largest proportion of fall-related admissions. Previous observational studies found that older women have a higher likelihood of falls than older men [12–14]. Gale et al. [15] sought to investigate sex-specific associations between potential risk factors and the likelihood of falling in patients aged 60 and older, in England. This cross-sectional study suggested that there were differences between the sexes in some risk factors for falls, such as incontinence and frailty in women, and older age, higher symptoms of depression and being unable to perform a standing balance test in men. In a subsequent longitudinal study, Gale et al. [16] examined the relationship between a wide range of factors and the risk of incident falls. The results suggest that the association between pain, balance and comorbidity and incident falls risk varied by sex. A longitudinal study by Ek et al. [17] also suggests that men and women have different fall risk profiles.

Areas of residence with a higher aging index presented the highest rates of fall-related admissions, with the Centre region of Portugal registering the highest rates over the years.

**Table 4. Number of fall-related admissions by type of fall, in the Portuguese population.**

| | 2010 | 2011 | 2012 | 2013 | 2014 | 2015 | 2016 | 2017 | 2018 |
|---|---|---|---|---|---|---|---|---|---|
| **Admissions** | **38,288** | **43,592** | **41,578** | **44,173** | **43,788** | **45,117** | **47,193** | **42,042** | **37,245** |
| **Type of falls** | **n (%)** | **n (%)** | **n (%)** | **n (%)** | **n (%)** | **n (%)** | **n (%)** | **n (%)** | **n (%)** |
| Accidental fall on or from stairs or steps | 1088 (2.84) | 1181 (2.71) | 1109 (2.67) | 1378 (3.12) | 1354 (3.09) | 1460 (3.24) | 1452 (3.08) | 1571 (3.74) | 1318 (3.54) |
| Accidental fall on or from ladders or scaffolding | 144 (0.38) | 185 (0.42) | 150 (0.36) | 266 (0.60) | 297 (0.68) | 310 (0.69) | 324 (0.69) | 375 (0.89) | 390 (1.05) |
| Accidental fall from or out of building or other structure | 580 (1.51) | 596 (1.37) | 567 (1.36) | 503 (1.14) | 468 (1.07) | 409 (0.91) | 413 (0.88) | 435 (1.03) | 381 (1.02) |
| Accidental fall into hole or other opening in surface | 71 (0.19) | 52 (0.12) | 64 (0.15) | 88 (0.20) | 54 (0.12) | 66 (0.15) | 82 (0.17) | 100 (0.24) | 103 (0.28) |
| Other accidental falls from one level to another | 2701 (7.05) | 2689 (6.17) | 2368 (5.70) | 2797 (6.33) | 2707 (6.18) | 2936 (6.51) | 2851 (6.04) | 3053 (7.26) | 2756 (7.40) |
| Accidental fall on same level from slipping tripping or stumbling | 3832 (10.01) | 3596 (8.25) | 3038 (7.31) | 2564 (5.80) | 2468 (5.64) | 2326 (5.16) | 2690 (5.70) | 3426 (8.15) | 2121 (5.69) |
| Fall on same level from collision, pushing, or shoving, by or with other person | 474 (1.24) | 859 (1.97) | 953 (2.29) | 1006 (2.28) | 682 (1.56) | 368 (0.82) | 325 (0.69) | 96 (0.23)* | - |
| Fracture, cause unspecified | 1550 (4.05) | 1574 (3.61) | 1385 (3.33) | 1521 (3.44) | 2000 (4.57) | 2635 (5.84) | 3586 (7.60) | 15 (0.04)* | - |
| Other and unspecified fall | 27,381 (71.51) | 32,141 (73.73) | 31,417 (75.56) | 33,636 (76.15) | 33,411 (76.30) | 34,333 (76.10) | 35,235 (74.66) | 32,797 (78.01) | 29,879 (80.22) |
| Late effects of accidental fall | 541 (1.41) | 802 (1.84) | 592 (1.42) | 471 (1.07) | 405 (0.92) | 365 (0.81) | 327 (0.69) | 251 (0.60) | 268 (0.72) |

*Type of fall only included in ICD-9, and no matching code in ICD-10.

This difference highlights the importance of age as a risk factor for falls. In a recent study by Chang et al., [18] falls were considered as one of the 92 age-related diseases (diseases with incidence rates among the adult population increasing quadratically with age).

Across all ages, the most common type of fall was "Other and unspecified fall" followed by "Accidental fall on same level from slipping tripping or stumbling". Patients aged 65 and older represented over 60% of the total sample. Kusljic et al. [19] examined the types of falls in a sample of 250 patients above the age of 65. They reported the most common type of falls being a fall on the same level related to slipping, tripping, or losing balance (35% of the sample), followed by unspecified fall on the same level (31%) and other types of fall (19%). Paul et al. [20] examined ambulance records and hospital admission data in New South Wales, Australia, among individuals aged 65 years and older and found that slips and trips were the most common mechanism of falls requiring hospitalization in this population (52%).

Different approaches to estimating fall-related hospital admissions rates resulted in estimates between 353 and 414 per 100,000 population. Identifying fall-related admissions based on episodes provides an overall picture of the landscape of falls in a scarcely explored setting in Portugal. However, such an approach poses limitations as to the uncertainty surrounding the true number of falls. This is because one event of fall may generate several hospital admissions. The approach based on the number of patients may underestimate the true number of falls because several events may be accrued to the same individual. Yet, identifying fall-related admissions based on episodes with a principal diagnosis of injury may increase the likelihood of selecting episodes of falls occurring outside the hospital setting and indicating that the injury was the chief reason for admission. A large proportion of fall-related admissions had a principal diagnosis of injury (about 85%), which may indicate that the fall was in fact the cause for hospital admission. The choice of which approach to take may depend on the aim of the analysis.

Adequately identifying what population groups may be at higher risk of falls and related morbidity is key to devising appropriate public health policies and programmes. The finding that higher rates of falls occur at different ages among men and women highlight the importance of gender-adequate preventive programmes, as also stressed by Wei et al. [12] and Ek et al. [17]. The analysis of type of fall suggest that further investigations are warranted to try to disentangle the aspects behind the classification of "Other and unspecified fall", which represented the largest proportion of fall-related admissions.Slipping, tripping, or stumbling as mechanisms of falls were the next most common type of fall (although relatively small proportion), which supports the importance of community-based intervention focusing on fall prevention. Further studies should examine sex differences underlying different mechanisms of falls.

The most recent GBD study [2] points out the complex multifactorial context of falls in the elderly and its most often iatrogenic underpinnings stemming from incorrect diagnosis and treatments. The study highlights the link between falls and psychotropic and cardiovascular medications, cognitive impairment, depression, and general frailty. Extensive evidence exists on the effectiveness of multifactorial interventions combining education, exercise, and interventions targeting home safety modification [21], and international guidelines exist for the assessment and prevention of falls in the elderly [22].

The global report on falls prevention, issued by the World Health Organisation in 2007 [23], discussed the importance of population ageing in the epidemiology and burden of falls. The report highlights the need for knowledge and preparedness of primary health care and social service providers on the complexity of factors that predispose falls in the older age. This is critical to ensure appropriate treatment and management of falls within primary care, as well as access to falls prevention programmes. At the national level, a report issued by the Portuguese Directorate-General of Health [24] drafted a national strategy for healthy ageing and reiterates the importance of investing in the creation of safe physical environments for the prevention of falls and related injuries. Other international agencies [25] followed suit and have stated the importance of investment in healthy ageing and building inclusive societies recognising its importance for overall population wellbeing and economic growth.

This study provides an overall picture of the landscape of falls in a scarcely explored setting in Portugal. Additionaly, the database used to describe the distribution of falls covers about 70% of all inpatient hospital admissions to public hospitals in the country. There are, however, a few limitations to take into account. Our study only accounts for one of the dimensions included in the injury pyramid for falls [26, 27]. This pyramid considers four levels of injury including general practitioner registry, emergency department registers, hospital discharge and mortality data. The effect of falls in general practitioner or emergency department visits (without inpatient admission) and deaths (which occurred outside the hospital setting) is not included in our estimates as the data pertains only to the hospital setting.

The transition from ICD-9-CM to ICD-10-CM/PCS during the study period (from 2016 onwards) may have had an impact on the coding of external causes of injury. It is not possible to ascertain the impact of such transition in our results.

The Hospital Morbidity Database used in this study to describe the landscape of fall-related admissions in Portugal has some limitations. This database was created to monitor hospital productivity for mainland Portugal within the publicly financed NHS. The islands of Madeira and Azores have their own health subsystem and reporting, therefore they are underrepresented within the database. The rates reported may therefore better represent the setting of mainland Portugal.

## Conclusions

This study reveals that fall-related hospital admissions are higher among the younger and the oldest age groups, among males until the age of 60, females from the age of 60, and in areas of residence with a higher aging index. Annual rates of falls differed based on the methodological approach chosen, which warrants further study. Further research should be conducted to better understand the risk profiles of these population groups in the Portuguese setting, and help design and implement appropriate policy for prevention of falls in at risk groups.

## Supporting information

**S1 Table. Distribution of rates of falls based on patients and admissions related to falls, between 2010 and 2018, in the Portuguese population.**
(DOCX)

**S2 Table. Distribution of number of patients and inpatient admissions related to falls, between 2010 and 2018, in the Portuguese population.**
(DOCX)

## Author Contributions

**Conceptualization:** Filipa Sampaio, Raquel Ascenção, Andreia Costa.

**Data curation:** Paulo Nogueira.

**Formal analysis:** Paulo Nogueira.

**Investigation:** Filipa Sampaio, Raquel Ascenção, Andreia Costa.

**Methodology:** Filipa Sampaio, Paulo Nogueira, Raquel Ascenção, Andreia Costa.

**Writing – original draft:** Filipa Sampaio, Raquel Ascenção.

**Writing – review & editing:** Paulo Nogueira, Adriana Henriques, Andreia Costa.

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
