## [Decision Letter · Decision Letter 0]

6 Sep 2021

PONE-D-21-21269The epidemiology of falls in Portugal: an analysis of hospital admission dataPLOS ONE

Dear Dr. Sampaio,

Thank you for submitting your manuscript to PLOS ONE. After careful consideration, we feel that it has merit but does not fully meet PLOS ONE’s publication criteria as it currently stands. Therefore, we invite you to submit a revised version of the manuscript that addresses the points raised during the review process.

We look forward to receiving your revised manuscript.

Kind regards,

Osama Farouk

Academic Editor

PLOS ONE

“ This work was produced with the computational support of INCD – Instituto Nacional de

Computação Distribuida – which is funded by FCT and FEDER under the project

01/SAICT/2016 No. 022153”

 “This publication was supported by Fundação para a Ciência e Tecnologia (FCT) under the references UIDB/04295/2020 and UIDP/04295/2020.”

“This publication was supported by Fundação para a Ciência e Tecnologia (FCT) under the references UIDB/04295/2020 and UIDP/04295/2020.”  

Reviewers' comments:

Reviewer's Responses to Questions

**Comments to the Author**

1. Is the manuscript technically sound, and do the data support the conclusions?

Reviewer #1: Partly

2. Has the statistical analysis been performed appropriately and rigorously? 

Reviewer #1: No

3. Have the authors made all data underlying the findings in their manuscript fully available?

Reviewer #1: No

4. Is the manuscript presented in an intelligible fashion and written in standard English?

Reviewer #1: No

5. Review Comments to the Author

Reviewer #1: General comments:

The study idea about the epidemiology of falls in the Portuguese population is interesting with this large sample size in nine years and is of great research importance.

But unfortunately, this work is not presented in a way that can deal with rich data. However, I have provided some remarks below.

Title:

“Epidemiology” is a wider term than what was studied and presented in the results.

Abstract:

The abstract all over its section should be corrected accordingly after rewriting of the manuscript.

Introduction

- Is well written

Methods:

- The study design was not included in this section

- The sentence in lines 112 and 113 is not related to the methodology, please delete.

- In page 12: lines 114 to 128: write the idea in two sentences only. Go direct to the data source and case identification, this section is informative and needed

- The available data related to falls in the “ Hospital Morbidity Database “ were not discussed in this section to be used in analysis.

- In statistical analysis section: there are no details about the analysis, The used variables should be written and use statistical significance tests

Results:

- In general:

• include “ in Portuguese population” to the titles of the tables and figures

• Overall, there is no tests of significance were used in analysis and to make comparisons within the groups in each table and to compare with the results of other studies in the discussion section. It’s important to do further analysis

• Classify the age groups in tables and figures in “ten years classes” of age in each group, and not by 5 years in this wide range of age., the last one could be 70+

• In tables including rates, add per 100,000 in the title

- In table (1):

• the numbers are written in two lines, this not an acceptable way to present numbers.

• add the “%” and “minimum -maximum” for presentation of data to be clearer.

• add the “SD” to mean number and do comparisons by the T test

- Figure (1):

• write (0-10) and not (0,10) in classes of age

- Page 15, line 183: correct to Figure (2)

- The distribution of falls according to demographics is not restricted to age and sex

- Table (2):

• add the “%” presentation of data to be clearer.

• replace “MF” by total

• aggregate the age classes as mentioned before

Discussion:

In general:

- The discussion is not well structured and not well written

- There are multiple undefined ideas

- There are redundancy with unimportant multiple sentences

- There are no significance in the presented data to compare the results with other studies or to compare within the same table e.g between age categories or between males and females “ on what basis???”

- Delete the title “ summary of main findings” and delete all this section to be rewritten.

- Limitations: shorten to 2-3 sentences and put them at the end of the discussion

- Methodology limitation: to not include mortality data and other data due to “the nature of the study” unclear??

- Add strength points

- “Policy implications and directions for future research”: this is not unclear section, is it a discussion or a conclusion, writing about the prevention as a recommendation with the conclusion in one sentence is better

- Conclusion:

Conclusion:

- shorten it and write about the main findings

- “This study provides…..” in the conclusion section, this could be added to the strength points not a conclusion.

6. PLOS authors have the option to publish the peer review history of their article (what does this mean?). If published, this will include your full peer review and any attached files.

Reviewer #1: **Yes: **Dalia G Mahran

---

## [Author Response · Author response to Decision Letter 0]

25 Oct 2021

Response to journal requirements and reviewers’ comments

We would like to thank the editor and the reviewer for their comments, which have significantly contributed to the paper. We have made changes in the paper according to their suggestions. All changes are highlighted by using the track changes mode in the main manuscript and presented under each comment in blue.

We have made sure the manuscript meets PLOS ONE´s requirements, including those for file naming. 

2. Thank you for stating the following in the Acknowledgments Section of your manuscript: “ This work was produced with the computational support of INCD – Instituto Nacional de Computação Distribuida – which is funded by FCT and FEDER under the project 01/SAICT/2016 No. 022153”. We note that you have provided additional information within the Acknowledgements Section that is not currently declared in your Funding Statement. Please note that funding information should not appear in the Acknowledgments section or other areas of your manuscript. We will only publish funding information present in the Funding Statement section of the online submission form. Please remove any funding-related text from the manuscript and let us know how you would like to update your Funding Statement. Currently, your Funding Statement reads as follows: “This publication was supported by Fundação para a Ciência e Tecnologia (FCT) under the references UIDB/04295/2020 and UIDP/04295/2020.” Please include your amended statements within your cover letter; we will change the online submission form on your behalf.

3. Thank you for stating the following financial disclosure: “This publication was supported by Fundação para a Ciência e Tecnologia (FCT) under the references UIDB/04295/2020 and UIDP/04295/2020.” Please state what role the funders took in the study. If the funders had no role, please state: "The funders had no role in study design, data collection and analysis, decision to publish, or preparation of the manuscript." If this statement is not correct you must amend it as needed. Please include this amended Role of Funder statement in your cover letter; we will change the online submission form on your behalf.

According to the editor´s comments 2 and 3 above, we have removed any funding information from the manuscript. The acknowledgment section was also removed. We would like to delete the current funding statement (“This publication was supported by Fundação para a Ciência e Tecnologia (FCT) under the references UIDB/04295/2020 and UIDP/04295/2020.”) since we did not receive funding from this source during the course of this study. This source of funding was initially included as the authors considered them for funding the open access at Plos One. However, the affiliation of the first and corresponding author, Filipa Sampaio, from Uppsala University Sweden, has an agreement for open access. Hence, naming “This publication was supported by Fundação para a Ciência e Tecnologia (FCT) under the references UIDB/04295/2020 and UIDP/04295/2020.” as source of funding is not anymore relevant.

We would like to replace the old funding statement with the following: 

“The author Paulo Nogueira received computational support of INCD – Instituto Nacional de Computação Distribuida – which is funded by FCT and FEDER under the project 01/SAICT/2016 No. 022153. The funders had no role in study design, data collection and analysis, decision to publish, or preparation of the manuscript.”

4. In your Data Availability statement, you have not specified where the minimal data set underlying the results described in your manuscript can be found. PLOS defines a study's minimal data set as the underlying data used to reach the conclusions drawn in the manuscript and any additional data required to replicate the reported study findings in their entirety. All PLOS journals require that the minimal data set be made fully available. For more information about our data policy, please see http://journals.plos.org/plosone/s/data-availability. Upon re-submitting your revised manuscript, please upload your study’s minimal underlying data set as either Supporting Information files or to a stable, public repository and include the relevant URLs, DOIs, or accession numbers within your revised cover letter. For a list of acceptable repositories, please see http://journals.plos.org/plosone/s/data-availability#loc-recommended-repositories. Any potentially identifying patient information must be fully anonymized. Important: If there are ethical or legal restrictions to sharing your data publicly, please explain these restrictions in detail. Please see our guidelines for more information on what we consider unacceptable restrictions to publicly sharing data: http://journals.plos.org/plosone/s/data-availability#loc-unacceptable-data-access-restrictions. Note that it is not acceptable for the authors to be the sole named individuals responsible for ensuring data access. We will update your Data Availability statement to reflect the information you provide in your cover letter.

We would like to update the data availability statement as follows: 

“Data availability statement

All available data are within the manuscript and its supporting information file. This study was based on the analysis of proprietary data, owned by the Portuguese Central Administration of the Health System, I.P. (ACSS). Aggregated data can be accessed via the Transparency Portal held by the Portuguese Ministry of Health, at the following link: https://transparencia.sns.gov.pt/explore/?sort=title&q=morbilidade+hospital.”

Reviewers' comments to author:

Reviewer #1

The study idea about the epidemiology of falls in the Portuguese population is interesting with this large sample size in nine years and is of great research importance. But unfortunately, this work is not presented in a way that can deal with rich data. However, I have provided some remarks below.

We would like to thank the reviewer for the valuable comments provided. We have, to the best of our knowledge, tried to incorporate all suggestions provided. We hope the manuscript is now at a state deemed acceptable for publication.

Title: 

“Epidemiology” is a wider term than what was studied and presented in the results.

Thank you for the comment regarding the relevance of the title for the scope of the paper. We understand the concept of epidemiology is quite broad. MacMahon and Pugh definition, updated by Greenland and Rothman (2008) (MacMahon and Pugh 1970 modified by Greenland and Rothman Chapter 3 Measures of Occurrence page 32 in Modern Epidemiology 3rd Edition K.Rothman, S. Greenland and T. Lash 2008 Lippincott Williams & Wilkins), presented epidemiology as the study of the distribution and determinants of disease frequency in human populations. We have in our paper attempted to describe the distribution of falls in the Portuguese population, using national population data from a national registry of inpatient and ambulatory care within the public sector. This database covers about 70% of all inpatient hospital admissions in the country. We have clarified in the title that the scope of the epidemiologic analysis pertains to hospital related data.

Abstract: 

The abstract all over its section should be corrected accordingly after rewriting of the manuscript.

We have adjusted the abstract according to the changes incorporated in the manuscript. 

Introduction: 

Is well written

Methods:

- The study design was not included in this section

The study design has been included at the beginning of the method section. Please see page 6, lines 135-136:

“This study used a descriptive design to examine the epidemiology of fall-related hospital admissions in the Portuguese population between 2010 and 2018.”

- The sentence in lines 112 and 113 is not related to the methodology, please delete.

The sentence has been deleted. 

- In page 12: lines 114 to 128: write the idea in two sentences only. Go direct to the data source and case identification, this section is informative and needed.

The section on study setting has been deleted and some of its information shortened and integrated in the section on “Data sources and case identification”. Please see pages 5 – 6, lines 115-152 for the new restructured section.

- The available data related to falls in the “ Hospital Morbidity Database “ were not discussed in this section to be used in analysis.

Thank you for pointing this out. We have below attempted at clarifying this matter, as well as have included in the manuscript more information under the Methods section on “Data sources and case identification”. Please see pages 7-8, lines 153-170.

Being primarily created for administrative purposes, the Hospital Morbidity Database comprises data on patient demographics, admission, and discharge information (including responsible hospital, date, and time), in-hospital mortality, International Classification of Diseases (ICD) diagnosis codes, and diagnostic related grouping codes (for cost information). 

The Hospital Morbidity Database is owned by the Portuguese Central Administration of the Health System, I.P. (ACSS) and its access is possible upon request for investigational purposes.

By using the Hospital Morbidity Database, we are limited to data collected at a national level, primarily for administrative purposes, where clinical information is coded by trained physicians (ICD). ICD-codes for external causes were used to identify the occurrence of fall and to categorize it according to the type of fall. ICD-10-CM provides further detail and granularity than ICD-9-CM and it was necessary to crosswalk between the two to present and interpret data. 

- In statistical analysis section: there are no details about the analysis, The used variables should be written and use statistical significance tests.

The Hospital Morbidity Database is a national database created for administrative purposes; therefore, all variables are routinely collected, independently from our study. This means that we were limited to the variables included in this database, as described in the previous comment. 

Thank you for your comment regarding the use of statistical tests. We would like to clarify our reasoning behind not using inferential statistics and thereby significance estimates. The database used in this study is a national database, used primarily for administrative purposes, which routinely collects data from the hospitals from the National Health Service. This data collection is done independently from our study. This means that this database contains inpatient and selected outpatient information for the Portuguese mainland population, which amounts to about 10 million people. Given these circumstances, this database provided us with the ideal setting to explore the landscape of falls, which had not yet been done for the Portuguese setting. Being aware of the many dimensions of falls, we were able to explore one of its dimensions pertaining to hospital admission-related falls. This was study was, hence, exploratory and aimed at investigating the distribution of falls in the whole inpatient population. Therefore, we had no a priori hypothesis to test, as this requires previous knowledge of the phenomenon, which we did not have. We would like to thank you for giving us the opportunity to clarify this matter in this reply, as well as in the manuscript. We have added a short sentence in the methods section, under “Statistical analyses”, on page 8, lines 173-175.

Other authors have done similar work of describing the distribution of events in different populations, without the use of inferential statistics, when on a population level. For instance, James et al (2017) with the Global Burden of Disease Collaboration has estimated the burden of falls in different countries (reference below).

James SL, Lucchesi LR, Bisignano C, et al The global burden of falls: global, regional and national estimates of morbidity and mortality from the Global Burden of Disease Study 2017

Injury Prevention 2020;26:i3-i11.

Results:

- In general:

• include “ in Portuguese population” to the titles of the tables and figures

This has now been included in all titles of tables and figures.

• Overall, there is no tests of significance were used in analysis and to make comparisons within the groups in each table and to compare with the results of other studies in the discussion section. It’s important to do further analysis

Thank you for your comment. Please refer to the previous comment on our reasoning behind the absence of inferential statistics and statistical tests. 

• Classify the age groups in tables and figures in “ten years classes” of age in each group, and not by 5 years in this wide range of age., the last one could be 70+

Thank you for your comment. We would like to clarify that the age group intervals used in our study are in agreement with the official age group intervals used by Statistics Portugal (https://www.ine.pt/xportal/xmain?xpid=INE&xpgid=ine_main), which is the national statistics authority. With this in mind, these age group intervals also allow our data and findings to be used by other researchers in the field who would like to have detailed information on the phenomenon of falls in Portugal. This is in line with the goal of the paper, that is to provide a detailed (as detailed as possible) overview of fall-related hospital admissions. 

• In tables including rates, add per 100,000 in the title

This has been added to all tables reporting rates.

- In table (1):

• the numbers are written in two lines, this not an acceptable way to present numbers.

• add the “%” and “minimum -maximum” for presentation of data to be clearer.

• add the “SD” to mean number and do comparisons by the T test

Table 1 has been corrected. Please refer to the previous comment on our reasoning behind the absence of inferential statistics and statistical tests. 

- Figure (1):

• write (0-10) and not (0,10) in classes of age

Figure 1 has been corrected. 

- Page 15, line 183: correct to Figure (2)

This has been corrected and the reference to Figure 1 has been replaced by Figure 2.

- The distribution of falls according to demographics is not restricted to age and sex

Thank you for your comment. We have corrected this section of results so that it reads as “Admissions related to falls by age and sex” (page 12, line 224)

- Table (2):

• add the “%” presentation of data to be clearer.

Table 2 has been corrected for ease of understanding. We have used a thousand separators for numbers related to the population. All rates are presented as rates per 100,000 population at risk.

• replace “MF” by total

This has been replaced by “Total”. 

• aggregate the age classes as mentioned before

Thank you for your comment. We would like to clarify that the age group intervals used in our study are in agreement with the official age group intervals used by Statistics Portugal (https://www.ine.pt/xportal/xmain?xpid=INE&xpgid=ine_main), which is the national statistics authority. With this in mind, these age group intervals also allow our data and findings to be used by other researchers in the field who would like to have detailed information on the phenomenon of falls in Portugal. This is in line with the goal of the paper, that is to provide a detailed (as detailed as possible) overview of fall-related hospital admissions. 

Discussion:

In general:

- The discussion is not well structured and not well written

- There are multiple undefined ideas

- There are redundancy with unimportant multiple sentences

- There are no significance in the presented data to compare the results with other studies or to compare within the same table e.g between age categories or between males and females “ on what basis???”

- Delete the title “ summary of main findings” and delete all this section to be rewritten.

- Limitations: shorten to 2-3 sentences and put them at the end of the discussion

Thank you for your comments on the discussion section of the paper. We have restructured and rewritten the discussion so it reads better and presents ideas in a more clear way. We have also shortened the text on limitations. Please see the full discussion section on pages 16-21.

- Methodology limitation: to not include mortality data and other data due to “the nature of the study” unclear??

Thank you for your question. The hospital morbidity database used in this paper includes data on ambulatory and inpatient care for Portuguese public hospitals (about 70% of all inpatient hospital admissions to public hospitals in the country). When studying falls, we are limited to hospital related data, hence we do not have access to the falls that occurred and were limited to community/primary care. Additionally, we do not have data on deaths due to falls in the community, nor does our data allow us to know if deaths occurring within the hospital setting are due to a fall. We have attempted to clarify this aspect in the text. 

- Add strength points

A couple of strengths point have been added. Please see page 19, lines 84-88.

- “Policy implications and directions for future research”: this is not unclear section, is it a discussion or a conclusion, writing about the prevention as a recommendation with the conclusion in one sentence is better

Thank you for your comment. We have restructured this section and the discussion as a whole so that it flows better. We have also added a short note about prevention in the conclusion as suggested. Please see the full discussion section on pages 16-21 and the conclusion on page 21-22, lines 143-150.

Conclusion:

- shorten it and write about the main findings

- “This study provides…..” in the conclusion section, this could be added to the strength points not a conclusion.

Thank you for your suggestions. We have restructured the conclusion accordingly. Please see the conclusion on page 21-22, lines 143-150.

---

## [Decision Letter · Decision Letter 1]

3 Dec 2021

The epidemiology of falls in Portugal: an analysis of hospital admission data

PONE-D-21-21269R1

Dear Dr. Sampaio,

We’re pleased to inform you that your manuscript has been judged scientifically suitable for publication and will be formally accepted for publication once it meets all outstanding technical requirements.

Kind regards,

Osama Farouk

Academic Editor

PLOS ONE

Additional Editor Comments (optional):

Reviewers' comments:

Reviewer's Responses to Questions

**Comments to the Author**

1. If the authors have adequately addressed your comments raised in a previous round of review and you feel that this manuscript is now acceptable for publication, you may indicate that here to bypass the “Comments to the Author” section, enter your conflict of interest statement in the “Confidential to Editor” section, and submit your "Accept" recommendation.

Reviewer #1: All comments have been addressed

2. Is the manuscript technically sound, and do the data support the conclusions?

Reviewer #1: Yes

3. Has the statistical analysis been performed appropriately and rigorously? 

Reviewer #1: Yes

4. Have the authors made all data underlying the findings in their manuscript fully available?

Reviewer #1: Yes

5. Is the manuscript presented in an intelligible fashion and written in standard English?

Reviewer #1: Yes

6. Review Comments to the Author

Reviewer #1: (No Response)

7. PLOS authors have the option to publish the peer review history of their article (what does this mean?). If published, this will include your full peer review and any attached files.

Reviewer #1: **Yes: **Dalia G Mahran

---

## [Editor Report · Acceptance letter]

13 Dec 2021

PONE-D-21-21269R1 

The epidemiology of falls in Portugal: an analysis of hospital admission data 

Dear Dr. Sampaio:

I'm pleased to inform you that your manuscript has been deemed suitable for publication in PLOS ONE. Congratulations! Your manuscript is now with our production department. 

Kind regards, 

on behalf of

Dr. Osama Farouk 

Academic Editor

PLOS ONE